# Transparency at the Source: Evaluating and Interpreting Language Models With Access to the True Distribution

**Jaap Jumelet**  **Willem Zuidema**
Institute for Logic, Language and Computation
University of Amsterdam
{j.w.d.jumelet, w.zuidema}@uva.nl

## Abstract

We present a setup for training, evaluating and interpreting neural language models, that uses artificial, language-like data. The data is generated using a massive probabilistic grammar (based on state-split PCFGs), that is itself derived from a large natural language corpus, but also provides us complete control over the generative process. We describe and release both grammar and corpus, and test for the naturalness of our generated data. This approach allows us to define closed-form expressions to efficiently compute exact lower bounds on obtainable perplexity using both causal and masked language modelling. Our results show striking differences between neural language modelling architectures and training objectives in how closely they allow approximating the lower bound on perplexity. Our approach also allows us to directly compare learned representations to symbolic rules in the underlying source. We experiment with various techniques for interpreting model behaviour and learning dynamics. With access to the underlying true source, our results show striking differences and outcomes in learning dynamics between different classes of words.[1]

## 1 Introduction

When we train a Language Model on large natural language corpora, we are in effect estimating a probability distribution over possible next tokens or masked tokens. The true distribution is unknown, so we cannot directly quantitatively measure how good our estimation is, or qualitatively assess whether our LM has discovered 'true' underlying patterns in the data. The best we can do is to measure perplexity on a new sample from the same unknown distribution, and compare that perplexity to the perplexity we obtain with other model architectures or training regimes (Brown et al., 1992),

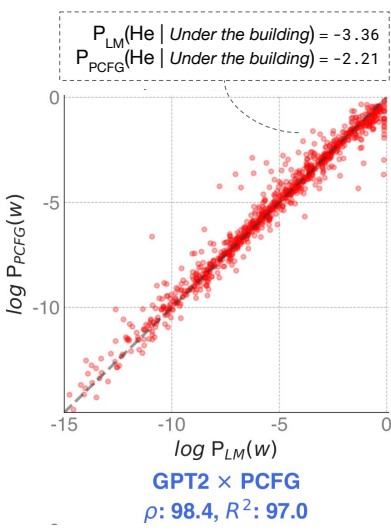

Figure 1: Spearman correlation between the log probabilities of a LM with GPT-2 architecture and the PCFG distribution. The LM has obtained a distribution that aligns significantly with the true PCFG distribution.

or with the expected perplexity given a (compute-optimal) scaling law (Kaplan et al., 2020; Hoffmann et al., 2022).

This approach has been enormously successful, but it leaves a number of interesting questions unanswered. First, one consequence of the absence of an explicit stochastic source is that it is impossible to exactly determine the *optimal perplexity* that a language model can obtain. Second, when designing interpretability methods to assess whether an LM has learned specific linguistic rules, we lack a *gold standard*. If an interpretability method fails to find evidence that the LM has learned a specific linguistic construction we cannot be sure whether this is a failure of the LM, a failure of the interpretability method, or whether our assumption that knowledge of this linguistic construction as an essential component of English fluency is wrong.

One approach to address these problems is to move to artificially generated data, but such data is often of trivial complexity compared to the richness

---

[1]All code, data, and checkpoints are available at: github.com/clclab/pcfg-lm.

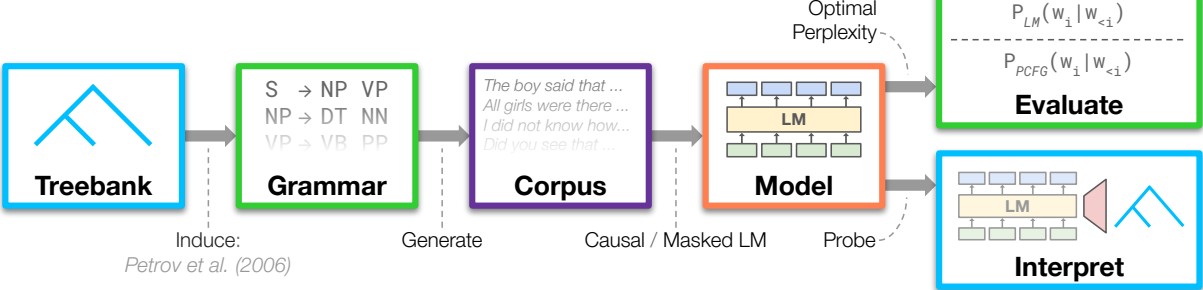

Figure 2: Conceptual overview of our experimental pipeline. First, we induce a massive probabilistic grammar from a natural language treebank. From this grammar we generate a corpus, that is used to train various language models on. Then, with access to the true distribution of our grammar, we evaluate and interpret our models.

of natural language, in terms of the vocabulary size or the number of grammatical rules.

In this paper, we explore an alternative approach. We also use generated data to train language models on, but to make sure the data approximates natural language in complexity, we use a *massive probabilistic grammar*, that is itself derived from a large natural language corpus. To obtain the grammar, we use an automatically parsed section from the The Pile corpus (Gao et al., 2020), and use the state-split framework (Petrov et al., 2006) – one of the most successful statistical parsing frameworks from before the rise of deep learning – to obtain a statistical grammar with more than 2 million rules. In §3.1 we describe the procedure for obtaining this grammar in detail.

This setup allows us to compute the exact lower bound on perplexity, although that computation turns out to still be nontrivial. This is due to the computational complexity of the problem that makes a naive approach infeasible, even on modern hardware. One key contribution from this paper is a closed-form expression to efficiently compute masked token probabilities for PCFGs, complementing the classic closed form for causal language modelling (Stolcke, 1995). Furthermore, our setup provides a gold standard to qualitatively assess results from interpretability methods against. We conduct a wide range of experiments, showing the naturalness of our generated data, determining the impact of model and data properties on language model performance, and interpreting model behaviour and learning dynamics using various interpretability tools. Figure 2 presents a schematic overview of our methodology.

## 2 LMs and Synthetic Data

Various approaches have controlled the structure of the training data to investigate particular properties of neural models or the learning process itself. A major line of work in this direction does this to investigate the impact of typological features of a language on model performance. For example, Cotterell et al. (2018) and Mielke et al. (2019) train LMs on parallel, translated corpora, which allows them to isolate the impact of typological differences between languages. Ravfogel et al. (2019) investigate typological differences as well, but create synthetic differences by modifying English sentences on typological features such as subject-object-verb order, an approach that has been pursued earlier by Wang and Eisner (2016). In all these papers the starting point of the training data is an (English) language corpus of which the underlying generative process is unknown.

There is a small set of papers where all data is derived from artificial languages, that are under full control. Dankers et al. (2022) generate naturalistic data for MT evaluation, but for models that were trained on natural data. Papadimitriou and Jurafsky (2023) investigate the inductive biases of LMs by pretraining models on data derived from simple grammars that isolate abstract linguistic properties such as nested recursion and Zipfian power laws. White and Cotterell (2021) provide a setup using simple PCFGs, in which typological features can be controlled. Hopkins (2022) observe that the simplistic nature of the PCFGs used in their work may not be reflective enough of natural language, and proposes Pitman-Yor processes to obtain artificial language that are more natural language-like. The grammars in these papers, however, remain fairly simple, and vocabulary size, sentence length and

other quantities stay limited compared to those seen in natural languages. Our setup provides a massive PCFG that is derived from natural language, with an efficient procedure for comparing the grammar's distributions to those of trained LMs.

## 3 PCFGs and Perplexity

In our work we make use of large-scale PCFGs to generate data on which we train various language models. In this section we give a brief background on PCFGs and how to evaluate modelling performance on PCFG-generated data using perplexity.

### 3.1 State-split PCFGs

There is a rich history in NLP of modelling language using interpretable, hierarchical grammatical formalisms. One of the most common formalisms is the Probabilistic Context-Free Grammar (PCFG), which provided the backbone for various successful parsing methods (Klein and Manning, 2003; Collins, 2003; Charniak and Johnson, 2005). A PCFG consists of simple weighted rewriting rules and can be used for both parsing and generation, by sampling and expanding rules based on the distribution described by the PCFG.

The context-freeness assumption of PCFGs, however, makes it challenging to capture non-syntactic dependencies between phrases.[2] One successful approach that addresses this is the **state splitting** procedure of Petrov et al. (2006), colloquially known as the '*Berkeley Parser*'. The general idea of state splitting is that we can split the non-terminal symbols in a PCFG into fine-grained sub-symbols, that are specialised towards specific linguistic phenomena using automatic learning. The splitting process is repeated multiple times, in an alternating split-and-merge strategy: if the splitting of a symbol does not result in a considerable improvement of the data likelihood the subsymbols are merged back. This results in an optimal trade-off between fine-grained non-terminal categories that still generalise beyond the training data. Furthermore, by keeping track of the history of split states we can project the final, fine-grained grammar back to the simpler, coarser base grammar. This then allows for highly efficient *coarse-to-fine* parsing, making it possible to parse sentences with large-scale grammars which would be intractable with a normal PCFG (Petrov and Klein, 2007).

### 3.2 Perplexity Lower Bound

The performance of a language model is often evaluated in terms of its **perplexity** on a test set (Jelinek et al., 1977; Jurafsky and Martin, 2009). Perplexity is defined as the normalised inverse probability of test set $\mathcal{D}$:

$$PPL(\mathcal{D}) = P(\mathcal{D})^{\frac{1}{|\mathcal{D}|}}$$
$$= \exp\left(\frac{\ln P(\mathcal{D})}{|\mathcal{D}|}\right)$$

Since the 'true' distribution over natural language is unknown, there is no way of determining the optimal perplexity that is attainable by a model.[3] In our setup, however, we *do* have access to the generative process behind our data. It therefore becomes possible to compute an exact lower bound for the perplexity, allowing us to express exactly how accurately a LM modelled the distribution set out by the PCFG.

The perplexity lower bound depends on the language modelling objective of interest: for masked language modelling it will be different than for causal language modelling. For both objectives we need to define a way to extract the required probability distributions out of the PCFG. We describe how this is possible in the next two sections.

### 3.2.1 Causal Language Modelling[4]

When doing causal language modelling, we aim to model the distribution $P(w_i|w_{<i})$: the probability of token $w_i$ conditioned on its prior context. The perplexity of a sentence can directly be decomposed into a product of such conditional probabilities, by continuously expanding the joint distribution over the entire string:

$$PPL(\mathbf{w}) = \left(\prod_i P(w_i|w_{<i})\right)^{\frac{1}{|\mathbf{w}|}}$$

Extracting a token distribution from a PCFG $G$, $P_G(w_i|w_{<i})$, is a topic that has been explored in detail since the 1990s (Jelinek and Lafferty, 1991; Hale, 2001). An efficient procedure is that of Stolcke (1995), who adapts the Earley parsing algorithm to compute the probabilities of prefix strings $w_{1i}$. The conditional token probability can then easily be recovered as the fraction of two subsequent prefix strings: $P_G(w_i|w_{<i}) =$

---

[2]For example, a simple PCFG may encode that *dog* is a noun and *bark* a verb, but from a single S→NP VP rule it can not be deduced that *dogs* are more likely to *bark* than *cats*.

[3]Shannon (1951) famously provides an approximation of the perplexity of the English language on a character level, but this does not yield an exact lower bound.

[4]Also known as *autoregressive* or *left-to-right* language modelling; the best fitting term remains a topic of debate.

$P_G(w_{1i})/P_G(w_{1(i-1)})$. In our experiments we make use of the implementation of Stolcke's procedure by Luong et al. (2013), called `EarleyX`.

### 3.2.2 Masked Language Modelling

When masked language modelling, we aim to approximate the probability distribution of $P(w_i|w_{\setminus i})$, which is achieved by masking out the token at position $i$ with a special mask token. Unlike in causal language modelling, a sentence's probability can not be decomposed into masked token probabilities. To aggregate the performance of a masked LM, we therefore need to make use of the pseudo-log-likelihood ($PLL$) (Salazar et al., 2020). The pseudo-log-likelihood expresses the log probability of a sentence as a sum of masked token probabilities:

$$\psi\text{-}LL(\mathbf{w}) = \sum_i \ln P(w_i|w_{\setminus i})$$

We can then use the $PLL$ to compute the *pseudo-perplexity* as follows:

$$\psi\text{-}PPL(\mathbf{w}) = \exp\left(\frac{\psi\text{-}LL(\mathbf{w})}{|\mathbf{w}|}\right)$$

To compute this quantity for a PCFG we need to find a way to extract masked token probabilities from the grammar: $P_G(w_i|w_{\setminus i})$. Since such a procedure does not yet exist, we propose a novel method.[5] Fortunately, it turns out to be possible to do this efficiently by employing the **inside-outside** algorithm (Baker, 1979; Manning and Schütze, 2001).

**Inside-Outside** The algorithm defines two probability distributions. The **inside** probabilities $\beta$ define the probability of a substring $w_{pq}$, given that the substring has non-terminal $N^j$ as its parent:

$$\beta_j(p, q) = P_G(w_{pq}|N^j_{pq}) \qquad (1)$$

The **outside** probabilities $\alpha$ define the joint probability of generating the non-terminal $N^j$ that spans position $p$ to $q$, as well as all the words outside the substring $w_{pq}$:

$$\alpha_j(p, q) = P_G(w_{1(p-1)}, N^j_{pq}, w_{(q+1)m}) \quad (2)$$

Using the in- and outside probabilities, we can directly express the masked token probability in a closed-form as follows:

$$P_G(w_i|w_{\setminus i}) = \sum_j \beta_j(i, i) \cdot \frac{\alpha_j(i, i)}{\sum_k \alpha_k(i, i)} \qquad (3)$$

We provide a detailed derivation in Appendix A. Our formulation allows for a highly efficient procedure of extracting these probabilities from a PCFG, since we can make use of the optimised coarse-to-fine parsing methodology of Petrov and Klein (2007) to obtain the in- and outside probabilities.

## 4 PCFG Corpus

Our approach can roughly be divided into three parts (Figure 2): i) **data generation** using state-split PCFGs, ii) **language modelling** using Transformer LMs, and iii) **model evaluation**, via probing-based interpretability tasks with a focus on *learning dynamics*. In this section we explain the data generation procedure, and do an analysis of our generated corpus.

### 4.1 Data Generation

**Grammar** To generate data we make use of the state-split PCFG procedure of Petrov et al. (2006)[6], as explained in §3.1. The treebank we use for learning a PCFG is a parsed version of the BookCorpus (Zhu et al., 2015; Conneau et al., 2018), which was parsed using the Stanford Parser (Manning et al., 2014), and is nowadays part of The Pile corpus under MIT License (Gao et al., 2020). We learn the PCFG on a sample of 500,000 sentences, which is the maximum corpus size we could learn on our system with 32GB RAM memory. Learning is done for 5 split-merge cycles, filtering rules with a probability below $10^{-8}$. Part of the learning procedure is transforming the base grammar to a X-bar style grammar, which ensures that each rule has at most two children.

The resulting PCFG consists of 54,497 unique tokens and 2.5M rules: 2.22M terminal rules, 272k binary non-terminal rules, and 13k unary non-terminal rules. The original treebank contained 96 non-terminal symbols: after split-merging we obtain 718 split non-terminals. We plot a fine-grained plot of the number of times each of the original 96 non-terminals has been split in Appendix B.

**Data** Having learned the PCFG from a treebank, we can sample from it to generate our training corpus data. For training our models we generate 7.5 million sentences (97 million tokens), and development/test/evaluation corpora of 50k/50k/10k sentences, respectively. Sentence lengths are capped

---

[5]Concurrent to our work, Zhao et al. (2023) find a similar formulation for expressing masked token PCFG probabilities.

[6]https://github.com/slavpetrov/berkeleyparser

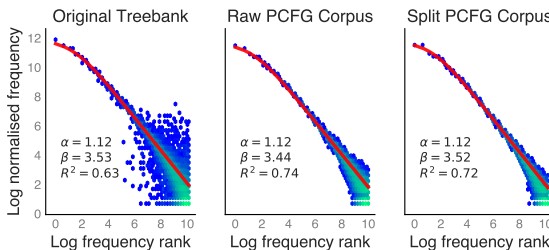

Figure 3: Relationship between a token's frequency rank and its frequency, which is near linear on a log-log plot. Rank and frequency are computed over two disjoint splits of the corpora, following the procedure of Piantadosi (2014).

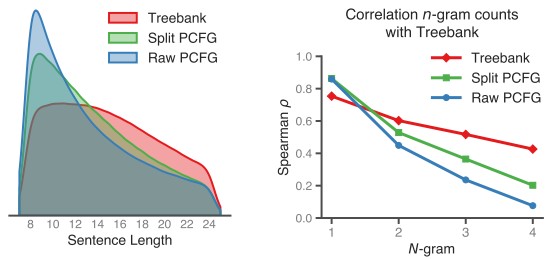

Figure 4: Distribution over sentence length for the three corpora, and the correlation of $n$-gram distributions with respect to the original Treebank data.

between 6 and 25, and duplicate sentences are allowed (but with no overlapping sentences between the train/development/test corpora): if we would not allow this, the data distribution that the model has access to would not reflect the original PCFG distribution. We provide a more detailed analysis of the final grammar and corpora in the following section, and a sample of sentences in Appendix C.

## 4.2 Naturalness of PCFG Corpus

In this subsection we briefly investigate whether the corpus generated from our grammar follows similar patterns as natural language. We compare the properties of our fine-grained PCFG to both the original Treebank, and a simple count-based PCFG that is derived directly from the original Treebank distribution. We refer to these three sources as *Split PCFG*, *Treebank*, and *Raw PCFG*, respectively.

**Zipf's Law**    One important linguistic feature that has been studied in detail within quantitative linguistics is that the frequency of a word is logarithmically proportional to its frequency rank. This feature is referred to as *Zipf's law*, and has been shown to be a universal language feature (Zipf, 1936). Zipf's law, in the updated form of Mandelbrot (1953), states that the $r^{\text{th}}$ most common word in a corpus has a frequency that is proportional to

$$f(r) \propto \frac{1}{(r + \beta)^\alpha} \quad (4)$$

We can investigate this feature for our generated data, and compare this against the frequency distribution of the original treebank. We follow the procedure of Piantadosi (2014), who argues that frequency and rank should not be computed on the same corpus, since that way a token's rank is always spuriously correlated to its frequency. Instead, we can compute the rank and frequency on

two independent corpora, and investigate the *Zipfian* relationship based on that. We estimate the $\alpha$ and $\beta$ parameters of Eq. 4 using the MLE implementation of Vogelmann (2020).

We plot our results in Figure 3. It can be seen that all three corpora follow Zipf's law with almost equal parameterisation of $\alpha$ and $\beta$. Interestingly, the generated corpora yield considerably lower residuals compared to the original treebank data. This demonstrates that in a natural language for infrequent tokens there exists greater uncertainty between the relation of frequency rank and frequency. Nevertheless, it is encouraging to see that our generated corpus follows a similar Zipfian distribution, likely an important property for the learnability of language (Kurumada et al., 2013; Hendrickson and Perfors, 2019).

**Sentence Length**    We investigate the distribution over sentence lengths in the corpora, and plot the histograms in Figure 4a. Both the PCFG corpora are skewed towards shorter sentences, whereas the original Treebank data is more uniformly spread out. The *Split PCFG* is more similar to the Treebank distribution than the *Raw PCFG*. We could opt to to subsample from our *Split PCFG* corpus to obtain a similar sentence length distribution as the Treebank corpus, but in that case the corpus distribution would no longer align directly with the PCFG itself.

**N-grams**    One aspect of natural language that state-split PCFGs aim to cover is that dependencies between words are often semantically correlated, as explained in §3.1. This phenomenon is known as *selectional preference*, and has been used by Hopkins (2022) to investigate the naturalness of their training data. There are many ways that selectional preference can be investigated, but we focus on a simple heuristic here. For increasing $n$

we compute the Spearman correlation between a corpus' frequency distribution over $n$-grams and the distribution of the original Treebank data. A high correlation then indicates that the corpus contains $n$-grams of a similar (natural) structure, and a low correlation indicates that the original $n$-gram dependencies are less present in the data. We compute this also for the Treebank itself, by splitting the corpus in half.

The results are shown in Figure 4b. For $n = 1$, the PCFG corpora both obtain a correlation that is higher than the Treebank with itself. This may be related to the large residual that we saw in the Zipf distribution of Figure 3, which showed that the unigram distribution of the Treebank corpus contains more variance in the tail than the PCFG corpora. For $n > 1$, it can be seen that the *Split PCFG* obtains higher Spearman correlation than the *Raw PCFG*, which demonstrates it has improved on the modelling of selectional preference.

**Other Frameworks** It is encouraging to see that our generated data generally follows similar patterns as the original natural data. However, we would like to stress that our usage of state-split PCFGs is certainly not the only option to obtain a generative process that models natural language well. Other frameworks that could be explored in future work include non-parametric Bayesian methods (Teh, 2006; O'Donnell, 2015), data-oriented parsing (Bod et al., 2003; van Cranenburgh et al., 2016), or $n$-gram based models (Pauls and Klein, 2012).

## 5 Language Modelling

We now move on to our experiments on language modelling the PCFG corpus that has been defined in the previous section. The underlying PCFG of the data generation process allows us to place the LM's performance against an optimal baseline.

### 5.1 Metrics

We compute the perplexity scores on the evaluation corpus for both the language models and the PCFG, using the methods of §3.2.[7] We skip over tokens

---

[7]Because the EarleyX algorithm has not been optimised for coarse-to-fine parsing, we were unfortunately forced to compute the causal PCFG probabilities on a smaller subset. The results reported are on a sample of 1100 token probabilities. Concurrent to our work, more efficient prefix probability algorithms have been introduced by Nowak and Cotterell (2023) and Opedal et al. (2023); we leave the incorporation of these procedures open for future work.

that are not part of the model tokenizer (i.e. those mapped to `<unk>`), since the language models did not have access to the true distribution of the original token. Based on the log probabilities of the language model and those of the PCFG, we can also compute their correlation. This allows us to determine to what extent the distribution of the LM is aligned with the true data distribution of the PCFG. We report two metrics: Spearman's $\rho$, a nonparametric rank correlation, and the $R^2$ coefficient, which measures the proportion of variation in the PCFG probabilities that is predictable from the LM probabilities.

### 5.2 Model Architecture

To emulate the learning behaviour of well-known Transformer LMs, we train LMs with similar architectures. For masked language modelling we train BERT, RoBERTa, and DeBERTa architectures (Devlin et al., 2019; Liu et al., 2019; He et al., 2021) of a much smaller size, inspired by the BabyBERTa models of Huebner et al. (2021). For causal language modelling we train GPT-2 and OPT architectures (Radford et al., 2019; Zhang et al., 2022). In our experiments we use model architectures with the following configuration: 8 layers, 8 heads, 256-dimensional hidden states, a batch size of 64, and a learning rate of $5 \cdot 10^{-4}$. Although a more optimal configuration may be found through a more elaborate hyperparameter search, it already provides a compute-friendly setup for training highly adequate models, as we will see in subsequent experiments. For training the LMs we use the `transformers` library (Wolf et al., 2020), training the models for 1 epoch with a cosine learning rate scheduler. For tokenization we use a whitespace tokenizer, with a minimum token frequency of 5 in the training corpus (else it is mapped to an `<unk>` token), which results in a vocabulary of 23,921 tokens. We use the same tokenizer for all models, to ensure a fair comparison across models.

### 5.3 Language Model Performance

We report the perplexity scores for our language models and PCFG in Table 1. We provide a regression plot of the results for the GPT-2 model in Figure 1. The considerable difference in PCFG perplexity between masked and causal language modelling shows that the amount of possible continuations is, unsurprisingly, far greater for causal language modelling. Despite being a task with greater uncertainty, however, the causal LMs ac-

| | Architecture | Size | $(\psi)$-**PPL** | $\mathbf{R^2}$ | $\rho$ |
|---|---|---|---|---|---|
| *Masked* | PCFG | | 63.9 | | |
| | BERT | 9.5M | 71.9 | 92.7 | 96.9 |
| | RoBERTa | 9.5M | 71.8 | 92.7 | 96.9 |
| | DeBERTa | 10.7M | **71.1** | 93.0 | 97.0 |
| *Causal* | PCFG | | 183.1 | | |
| | GPT-2 | 12.7M | **192.8** | 97.0 | 98.4 |
| | OPT | 15.6M | 194.2 | 96.7 | 98.3 |
| *TB* | DeBERTa | | *41.1* | | |
| | GPT-2 | | *115.0* | | |

Table 1: Perplexity scores for various masked and causal LM architectures. $R^2$ and Spearman's $\rho$ are computed with respect to the PCFG log probabilities. Size denotes the total number of model parameters. *TB* denotes performance of LMs trained on the original treebank data.

quire distributions that are closer aligned to the true PCFG distribution than the masked LMs. It can be seen that a higher Spearman's $\rho$ and $R^2$ yields a lower perplexity for all models, which demonstrates that models improve their performance by aligning their distribution closer to the true distribution. As the DeBERTa and GPT-2 architectures obtain the best performance, we will use these architectures in subsequent experiments to evaluate and interpret their performance and behaviour.

**Treebank** We also trained masked and causal LMs on the sample of 500k sentences from the original treebank that had been used for inducing the state-split PCFG. This experiment serves as another baseline for the naturalness of the PCFG-generated data: the treebank data contains all possible linguistic cues that a LM could rely on, and a worsened performance on synthetic data indicates that some of these cues have been lost during grammar induction. We can see that this is the case in Table 1: the performance of both LMs is considerably better than the LMs trained on PCFG-generated data.

## 5.4 Corpus & Model Size

To investigate the impact of the amount of training data and the impact of model size, we conduct an additional experiment where we investigate how model performance shifts as these two aspects are varied. For the impact of corpus size we test models on increasing training sets: ranging from 10.000 to 7.5 million sentences. The model architectures is the same as outlined in §5.2, across all corpus

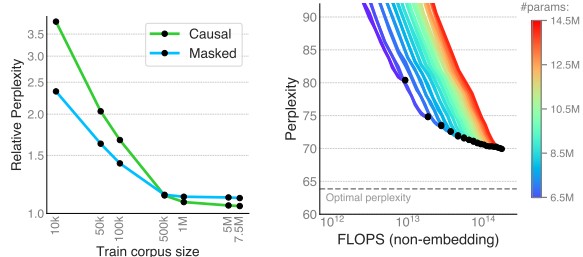

Figure 5: Language modelling performance expressed as a function of training size (a) and model size (b). Relative perplexity is computed with respect to the PCFG's perplexity lower bound, averaged over 3 seeds.

sizes. Each model is trained for the same amount of steps (15,000), but for smaller corpus sizes we set the maximum number of epochs to 50, to reduce overfitting. In order to compare causal and masked LM performance directly, we report the *relative perplexity*, which is defined as the LM perplexity divided by the PCFG perplexity lower bound.

For the impact of model size we train DeBERTa models with an increasing amount of layers, ranging from 1 to 20. All models are trained on the maximum corpus size of 7.5 million sentences for 1 epoch. We report model performance as a function of FLOPS across intermediate checkpoints, similar to how this is done for evaluating scaling laws of massive LMs (Kaplan et al., 2020; Hoffmann et al., 2022). FLOPS are approximated using the `fvcore` library.

**Results** We report the results in Figure 5. Interestingly, the causal LMs converge slower to the final performance than masked LMs, but ultimately reach a better relative perplexity. The corpus of 500,000 sentences appears to be the inflection point: the masked LMs appear to reach a plateau at this point, whereas the causal LMs continue to improve with more training data.

For the model size experiment it can be seen that performance improves as model size increases. A considerable gap between the optimal perplexity and model perplexity remains, however. The models we consider here are still relatively small, and it appears that larger scale models may be beneficial for the task, without risk of overfitting yet. Nonetheless, it remains interesting to see that our models follow a similar pattern as seen for large-scale LMs (Hoffmann et al., 2022), with a similar near-linear decrease in perplexity in log space.

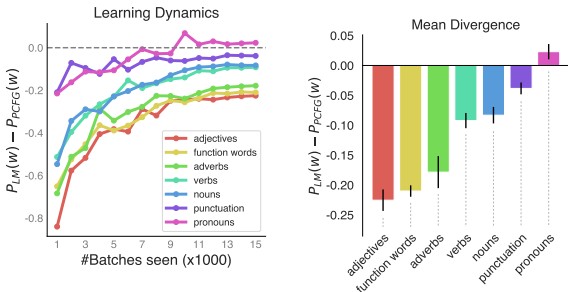

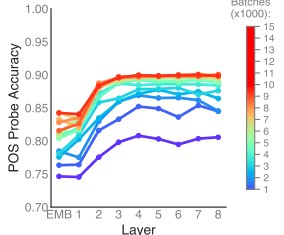

Figure 6: The LM probability divergence from the PCFG distribution, aggregated by general part-of-speech classes, for the DeBERTa model.

Figure 7: Left: POS Probing performance across layers (x-axis) and split out for multiple checkpoints through training. Right: Spearman correlation between POS probing probabilities of the true class and the token probabilities of the PCFG, LM, and probability divergence.

## 6 Model Interpretability

We investigate the behaviour and learning dynamics of the DeBERTa model from §5.3. Our setup provides an interesting test bed for interpretability purposes: since we have transparent access to the underlying structure of our data, we can make stronger assumptions about the cues the model must be relying on than would be possible in natural language. We conduct two experiments, one on learning dynamics and another on Part-of-Speech probing.

### 6.1 Learning Dynamics

The LMs of §5.3 obtain a distribution that aligns closely to the true PCFG distribution. We investigate how this distribution forms during training by aggregating token probabilities to a general POS class, mapping the 43 POS tags in the original data to one of 7 base classes. We report the *divergence* of the LM probabilities from the PCFG, expressed as the mean difference in log probabilities.

**Results** We report the results in Figure 6. We can see that different POS classes are learned at different stages during training. Punctuation and pronouns obtain a low divergence at the start of training already, which is maintained across checkpoints. A possible explanation for this might be that punctuation is less dependent of context, and more on the position in the sentence: quotation marks occur mostly at the start and end of the sentence, and most sentences end with a period symbol.

### 6.2 POS Probing

Probing is an interpretability technique in which auxiliary classifiers are trained on top of the representations of a LM, to find out whether abstract features are encoded (Adi et al., 2017; Hupkes et al.,

2018). We conduct a probing experiment on the representations of the DeBERTa model, classifying the part-of-speech (or pre-terminal) node above each token in the syntax tree.[8] We split the train and test sections based on the underlying token of a POS tag: this ensures that the probe finds a robust signal that does not simply map a token's identity to its most common POS tag, but that generalises based on other tokens with that POS tag (Hewitt and Liang, 2019).

**Results** We report the results in Figure 7. Performance reaches a plateau at 40% of training, after which it settles at an accuracy of around 90%. We can see that POS information is encoded consistently from around layer 3 in the model, and that this performance does not deteriorate in upper layers, unlike pretrained LMs like BERT (Tenney et al., 2019), where upper layers appear to encode sentence-level semantic features. This indicates that the PCFG nature of our data relies more on syntactic cues, which is likely the effect of a loss of *naturalness*, as explored in §4.2.

Additionally, we also investigate the correlation between the probabilities of the POS probe and the data distribution. We would expect that if a model relies on the POS signal extracted by our probe for its token predictions, that in cases where these predictions diverge from the PCFG distribution, the POS performance is lower as well. We compute the Spearman correlation for both the LM and PCFG distributions, as well as the divergence. Both LM and PCFG distributions obtain a positive correlation: this shows that the POS probe performs better on tokens with a higher probability. The divergence

---

[8]Note that our procedure right now does not take structural ambiguity into account, we leave the incorporation of the full POS distribution open for future work.

metric, however, yields a *negative* correlation: this shows that when the LM's probabilities diverge more from the PCFG distribution, the POS probing performance drops as well. We can conclude from this that representing POS information is a prerequisite for competent language model performance.

# 7 Conclusions and Future Work

In this paper, we train language models on artificial data from an explicit, known source, but in a setup where that data approximates natural language data in richness and complexity. This approach allowed us to compute an exact lower bound on perplexity, and evaluate how well different LM architectures can approach this optimum. We observed striking differences in the learning dynamics between masked and causal language models, where the latter converge much more slowly, but ultimately approximate the optimum more closely. And we observe large differences between word types in how well the trained LMs approximate the true probabilities from the underlying source grammar.

Our proposed methodology for evaluating and interpreting LMs thus involves inducing a rich probabilistic grammar, and using that grammar to generate a language-like training sets for neural language models. It did become clear, however, that our grammar induction has led to a sacrifice of various linguistic cues, which is demonstrated by a lower selectional preference (§4.2) and higher perplexity with respect to the original treebank (§5.3). We leave the exploration of which linguistic cues exactly are lost open for future work, for example by evaluating the LMs on the BLiMP benchmark for linguistic phenomena (Warstadt et al., 2020).

We believe our methodology holds promises beyond the specific modelling choices we have made in this paper. In the future we hope to see studies using different grammar induction procedures and focusing on other interpretability techniques, including the ambitious efforts, under the label 'mechanistic interpretability', to reverse engineer the behaviour of LLMs. Furthermore, we believe our work will be of interest to work on NLG evaluation and uncertainty estimation, where having access to the true data distribution is important (Pimentel et al., 2023; Zhang et al., 2023; Giulianelli et al., 2023).

Perhaps the most important relevance of our work, and such alternative approaches, is in relation to a key open problem in interpretability research:

how to evaluate the faithfulness of attribution and probing techniques. Prior research has attempted to define controlled environments that provide a certain degree of certainty about *how* the model is solving a task. Examples include the training of models on small-scale artificial data to perfection (Hao, 2020; Jumelet and Zuidema, 2023), or by explicitly providing shortcuts that the model *must* rely on (Bastings et al., 2022; Ebert et al., 2023). Scaling this up to more realistic, natural data, however, remains an open problem. Our setup, with direct access to a natural-like distribution, now provides much stronger guarantees about what patterns in the data the model is relying on, and allows for a gradual increase of the complexity of the data distribution.

# 8 Limitations

Our work involves *language-like* artificial data. An fundamental limitation is the degree to which stochastic grammars can indeed generate data that is language-like in important respects. Although our analyses in §4.2 show that our data is much more language-like than the baseline we considered, it is also clear that data still differs from natural language distributions. More work is needed to assess which differences between artificial and natural language datasets affect the generalizability of findings on the former to the latter.

A more mundane limitation of our approach to generating data is that we rely on a large automatically parsed corpus to induce our probabilistic grammar. It is clear that the automatic parses are far from error-free; it could be that the quality of our generated data would improve if the quality of the automatic parses is improved, by better automatic methods or manual correction.

Our closed form equations for computing the lower bound on perplexity are exact, but when computing the lower bound for specific large dataset we make use of some standard heuristics from the coarse-to-fine parsing literature where some extremely low probability productions in long sentences might be pruned from the chart. We don't expect this to have noticable effects, but have not rigorously evaluated this assumption.

The implementation we use for computing the lower bound in the causal language modelling case, EarleyX, is not optimized for extremely large grammars, and does not offer coarse-to-fine pruning. It is therefore painfully slow, limiting the size of

grammars which we could use. To scale up our approach to larger datasets, grammars and language models, a new, optimized implementation would be required. The recent approaches of Nowak and Cotterell (2023) and Opedal et al. (2023) may provide a solution for this.

In our interpretability analyses, where we make use of knowledge of the exact rules of the source grammar, we use the gold standard derivation tree for each sentence, and – for computational reasons – do not recompute the entire parse forest. In effect, we thus ignore structural ambiguity in those analyses; we don't expect this to have noticable effects, but have not rigorously evaluated this assumption.

## Acknowledgements

The authors are grateful fpr the useful feedback provided by Charlotte Pouw, Marianne de Heer Kloots, and the anonymous reviewers.

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

## A   Masked Language Modelling in PCFGs

In this section we demonstrate how the inside and outside probabilities can be used to compute masked token probabilities.

Firstly, the masked token probability for $w_i$ can be expressed as the marginalisation over all non-terminal symbols $N_{ii}^j$ that are the parent of $w_i$:

$$P_G(w_i|w_{\setminus i}) = \sum_j P_G(w_i, N_{ii}^j|w_{\setminus i}) \quad (5)$$

which can be rewritten to

$$\sum_j P_G(w_i|N_{ii}^j, w_{\setminus i}) \cdot P_G(N_{ii}^j|w_{\setminus i}) \quad (6)$$

The first term is equal to $P_G(w_i|N_{ii}^j)$: due to the context-free nature of PCFGs the probability of token $w_i$ solely depends on its parent non-terminal token, and not on neighbouring tokens in the sentence. This term is therefore equal to the inside probability $\beta_j(i, i)$:

$$P_G(w_i|N_{ii}^j, w_{\setminus i}) = P_G(w_{ii}|N_{ii}^j) \quad (7)$$
$$= \beta_j(i, i) \quad (8)$$

The second term can be expressed in terms of outside probabilities. In the special case of a substring of length 1 ($w_{ii}$), the words outside of the substring are equivalent to the masked token context $w_{\setminus i}$:

$$\alpha_j(i, i) = P_G(N_{ii}^j, w_{\setminus i}) \quad (9)$$

We can expand this as

$$\alpha_j(i, i) = P_G(N_{ii}^j|w_{\setminus i}) \cdot P_G(w_{\setminus i}) \quad (10)$$

The first term, $P_G(N_{ii}^j|w_{\setminus i})$, is equal to the second term in Eq. 6. The second term, $P_G(w_{\setminus i})$, can be marginalised out over all non-terminals $N_{ii}^k$; i.e. over all potential non-terminal parents of token $w_i$. By doing this we are able to express $P_G(N_{ii}^j|w_{\setminus i})$ in terms of (normalised) outside probabilities:

$$P_G(w_{\setminus i}) = \sum_k P_G(w_{\setminus i}, N_{ii}^k) \quad (11)$$
$$= \sum_k \alpha_k(i, i) \quad (12)$$
$$P_G(N_{ii}^j|w_{\setminus i}) = \frac{\alpha_j(i, i)}{P_G(w_{\setminus i})} \quad (13)$$
$$= \frac{\alpha_j(i, i)}{\sum_k \alpha_k(i, i)} \quad (14)$$

Where Eq. 13 is a simple reordering of Eq. 10. Plugging this all in Eq. 6 we obtain the masked token probability distribution expressed in terms of inside-outside probabilities:

$$P_G(w_i|w_{\setminus i}) = \sum_j \beta_j(i, i) \cdot \frac{\alpha_j(i, i)}{\sum_k \alpha_k(i, i)} \quad (15)$$

## B  Number of Non-terminal Splits

We plot the number of times each non-terminal type has been split in our final grammar in Figure 8. The more often a rule is split, the more fine-grained its distribution can become. It can be seen that in general open-class, pre-terminals lead to more state splits, although common non-terminals such as `NP`, `VP`, and `S` have been split often as well.

## C  Sample of Corpus Sentences

A sample of sentences taken from the evaluation corpus. We provide an example of the parse tree of the first sentence in Figure 9.

*Her mouth wasn't very close .*

*It met Calvin's lips and closed the wind .*

*The human ceased me to be average .*

*The Captain arrived and went apt of some not to think .*

*Liz spotted us through Bonjour and Monday , traveling around each of Ryland against the doorway .*

*After my injured procedure , you shrugged , backstabbing into the wall .*

*Josh kept departing thirty hundred alibis no king .*

*As I separates or eventually killed the one top of a giant length , Travis left .*

*A general car was trespassed at tunnels on the image of the misplaced formation .*

*I think that does find a perfect visit to activity .*

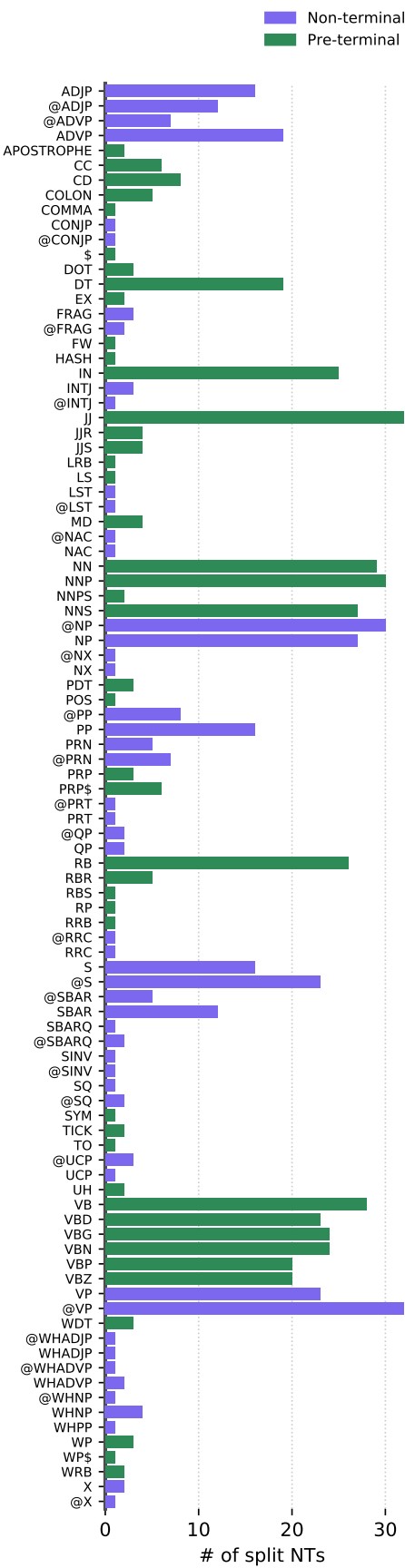

Figure 8: Number of splits per non-terminal. The '@' rules are the result of the X-Bar binarization procedure.

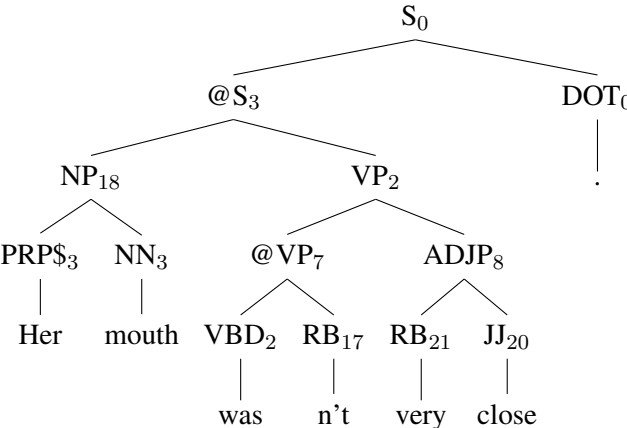

Figure 9: Example tree from the evaluation corpus. The @ non-terminals are the result of the X-Bar binarization procedure of Petrov et al. (2006).