# OpenReview forum: "Transparency at the Source: Evaluating and Interpreting Language Models With Access to the True Distribution"
_EMNLP/2023/Conference — EMNLP 2023 Findings_

### Official Review · Reviewer_ZP1G · 2023-07-31

**Soundness:** 2

**Excitement:**

3: Ambivalent: It has merits (e.g., it reports state-of-the-art results, the idea is nice), but there are key weaknesses (e.g., it describes incremental work), and it can significantly benefit from another round of revision. However, I won't object to accepting it if my co-reviewers champion it.

**Missing References:**

https://aclanthology.org/P12-1101.pdf (another approach to improving PCFGs)

**Paper Topic And Main Contributions:**

This paper trains LMs (both causal and non-causal/masked) on synthetic data generated from a PCFG. Unlike past work, their PCFG is large, giving it the capacity to be a better approximation to true English. Since LMs in this paper are trained from a known distribution, one can compare perplexities of the LMs to (approximations of) the lower bound from the ground truth model.

The authors find "striking" differences causal and masked LM objectives in their correlation with the true probabilities from the PCFG. They also study learning dynamics of word classes during training and find differences in the learning rates of classes of words (for example, punctuation is "learned" more quickly than adjectives).

RESPONSE to rebuttal:
> [W]e very much agree that the relevance does not come from PCFGs being particular good models of language, or from performance on PCFG-generated data being a good “sanity check”. Rather, the relevance of this work is, as we also wrote in response to Reviewer ybVx, in providing a distribution of intermediate complexity: not as language-like as those modeled by modern NLMs, but also not as idealized as in much prior work, and yet completely transparent.

This is a contradictor: you start by saying that it does not matter if PCFGs are particularly good models of language -- but then say that the value of this work is that they are "not as idealized" as much as prior work -- which in this context means "better models of language." This paper takes a formalism (PCFGs) with some nice interpretability characteristics and changes it on exactly one axis: how well it models real language. I do not see any justification for why "better but still bad" on this axis is useful.

These PCFGS are also very much *not* "transparent" -- the latent subcategories learned are just as opaque as the vectors in a modern LM, requiring their own probing to "interpret."

> We don't disagree! PCFGs are much worse approximations of NL; but the one aspect that still makes them interesting is their inherent interpretability.

I agree that PCFGs are more interpretable than neural LMs; it does not follow that neural LMs trained on PCFG-generated data are more "interpretable." I mentioned this conflation when I asked about this sentence:
> we can make stronger assumptions about the cues the model must be relying on than would be possible in natural language.

and do not see a response to it in your rebuttal.

Your answer to my question about
> We investigate how this distribution forms during training by aggregating token probabilities to a general POS class, mapping the 43 tags in the original data to one of 7 base classes.
did not answer what you did. I understand that the PCFG provides POS tags to generated sentences where other LMs do not. My question was how you compare the distribution of the LM -- which does not provide those POS tags -- with the PCFG that does. I can imagine several things you might have done but the text here is not clear. I should also note that the POS tags present during generation are not ground truth -- for each sentence, there are many trees (with many different POS tags) that could produce that sentence. It's probably the case the the POS tag posteriors given a sentence are fairly peaked, but I'm still

**Questions For The Authors:**

On line 542:
> We investigate how this distribution forms during training by aggregating token probabilities to a general POS class, mapping the 43 tags in the original data to one of 7 base classes.

What does this mean? It sounds like it means that each token is assigned to a fixed POS for each token in the vocabulary and you measure how well the learning LM matches the ground truth LM's distribution on that token type -- but POS tags are not fixed for the ground truth grammar? Why does the fact that the ground truth LM here is a PCFG matter? I think you could run this comparison on any ground truth generative model (like another causal LM)?

**Reasons To Accept:**

This paper does have one interesting result: I do not know why the correlation of the trained LM probabilities to the PCFG probabilities is so much higher for the causal objective. I think there's some chance this is an artifact of the obvious fact that causal probabilities are lower in general but I'm not sure. I wonder if KL divergence would have been a more informative metric here.

**Reasons To Reject:**

Unfortunately, I don't think the investigation in this paper is well motivated. Studying an LM architecture's ability to learn from artificial languages acts as a kind of sanity check -- can this model architecture even learn to approximate a particular class of models -- but that is not a contribution of this work? The contribution of this work is comparing learned LM perplexities to their lower bounds on a better -- but still not good -- PCFG approximation of English. This answers the question: how well can an LM architecture approximate a poor approximation of English? However, I do not think it says anything useful about what LM architecture does on real English, nor does it necessarily provide any more sanity checks than what already exist in the literature.

Taking a step back, a PCFG and a (causal autoregressive) LM are both generative models of English, but PCFGs are not inherently more interesting to study. In fact, modern causal LMs are far probably more interesting because they are _significantly_ better approximations of English. Note that the authors never show the perplexity other their PCFG on a dataset and compare it to a causal LM. I assume, based on the amount of data it was trained on and the samples in Appendix C, that it is not nearly as good as modern LMs. The authors show that it more closely approximates the tree bank than worse PCFGs, but not that it comes anywhere close to e.g. GPT2 as a model of English. (Side note: I also think the wording on line 410-11 "models natural language well" is not appropriate, since the authors have not shown that their PCFG models natural language "well.")

I think a PCFG is potentially more interesting for two reasons:
1) both the causal and masked loss functions can be computed efficiently -- or at least approximated efficiently. Computing the masked loss function p(w _i | w_{\I}) for a causal autoregressive LM is possible and not even that inefficient, though it would quite expensive to do for a whole corpus.
2) A PCFG has certain properties (like POS tags) that seem worth "interpreting."

Sacrificing LM quality for 1 seems pointless -- why compare your LM architecture's perplexity to a theoretical lower bound on a language that is not that much like English just because you can do so efficiently?

I think the authors are more interested in 2), but again, what lessons should I take from this? For example, the authors state (line 534)

> we can make stronger assumptions about the cues the model must be relying on than would be possible in natural language.

Is that true? "the model must be relying on" is a statement about how the model works, which is independent of the distribution of the data. I think there's some kind of identifiability argument to be made that if you manage to achieve the optimal perplexity with model A on a synthetic dataset generated from a model B, then you must be able to recover the underlying properties of B from A, but I don't think that's true for any particular probing strategy. There could still be "a failure of the interpretability method" to use the authors words. And in any case, the model architectures here fail to achieve the optimal perplexity, so who knows what they "must be relying on" to make their decisions.

The authors claim that their algorithm for computing P(w_i | w_{\I}) is a contribution. I think it is a fairly straightforward use of the probabilities from the inside-outside algorithm and can't be claimed as a contribution. For example, https://arxiv.org/abs/2303.08117 state it without much fanfare in equation (12). This was one of the top results when searching for "inside outside probabilities masked language modeling" on Google, though I understand this paper is from earlier this year and the authors may have missed it at time of writing, I would be surprised if this is the first time someone has computed this quantity.

Overall, I think I'd say that I don't know what a practitioner is supposed to have learned from this paper.

**Reproducibility:**

4: Could mostly reproduce the results, but there may be some variation because of sample variance or minor variations in their interpretation of the protocol or method.

**Reviewer Confidence:**

4: Quite sure. I tried to check the important points carefully. It's unlikely, though conceivable, that I missed something that should affect my ratings.

---

> ### Author Rebuttal · Authors · 2023-08-29
>
> We thank the reviewer for their review, and the interesting criticism on our work. We hope our reply, and the comments from the other reviewers, might lead to a more positive assessment of the potential relevance of our work!
>
> > Overall, I think I'd say that I don't know what a practitioner is supposed to have learned from this paper.
>
> The question here is who those practitioners are, in practice. For practitioners developing extremely large-scale LMs, our paper may not be of direct interest. Our target audience would rather be practitioners coming from an interpretability perspective and researchers with an interest in the linguistic properties of LMs. In interpretability research, the lack of faithfulness guarantees is a pressing issue, and a wide range of papers have resorted to using artificial language data to gain a more fundamental insight into LM behavior, as outlined in Section 2 of the paper.  We will try to sharpen these points further in the introduction, to make even clearer were we see the relevance of this work; we very much agree that the relevance does not come from PCFGs being particular good models of language, or from performance on PCFG-generated data being a good “sanity check”. Rather, the relevance of this work is, as we also wrote in response to Reviewer ybVx, in providing a distribution of intermediate complexity: not as language-like as those modeled by modern NLMs, but also not as idealized as in much prior work, and yet completely transparent.
>
> > Taking a step back, a PCFG and a (causal autoregressive) LM are both generative models of English, but PCFGs are not inherently more interesting to study. In fact, modern causal LMs are far probably more interesting because they are significantly better approximations of English.
>
> We don't disagree! PCFGs are much worse approximations of NL; but the one aspect that still makes them interesting is their inherent interpretability and links to linguistic theory. While parts of our experimental setup could be done using an auxiliary language model as a source model (e.g. GPT-2), this would not resolve the inherent issue of having full control and transparency over the source data.
>
> We were also surprised we couldn't find any prior work that had computed the masked LM probability for a PCFG, and are grateful for the referenced preprint which is indeed very relevant (for those authors, the closed form expression is the crucial step in proving theorem 1; in our work, it is crucial to adapt the Petrov et al. parser to compute the quantities we can compare with Transformer-based Masked LMs).
>
> Q1: Our data is generated following the PCFG rules. This means that a leaf node is generated from a rule of the form A -> w, and this final non-terminal A is what we’d call the POS tag of the token. If we were to use an auxiliary LM for generating the corpus, we would not have this knowledge, i.e. we wouldn’t know whether the auxiliary LM uses abstract POS classes for generating a token. This explicit linguistic structure is one of the reasons we want to use an interpretable PCFG instead of another black-box LM. We pose this is important, because numerous works on probing (e.g. Tenney et al., 2019a & 2019b) rest on the assumption that such linguistic abstractions are present in natural language, and that therefore it’s worth investigating whether LMs have picked up on these notions. Our setup provides explicit guarantees about the linguistic structure that a LM may pick up on.

---

### Official Review · Reviewer_45J7 · 2023-08-02

**Soundness:** 4

**Excitement:**

3: Ambivalent: It has merits (e.g., it reports state-of-the-art results, the idea is nice), but there are key weaknesses (e.g., it describes incremental work), and it can significantly benefit from another round of revision. However, I won't object to accepting it if my co-reviewers champion it.

**Paper Topic And Main Contributions:**

When training language models, we evaluate them according to the perplexity measure. But because the true distribution underlying natural language is unknown, we do not have access to the lower bound on perplexity.

This paper, instead, trains LMs on synthetic data generated using a state-split PCFG obtained using automatically parsed data from The Pile. This allows computing an exact lower bound of the perplexity and provides better interpretability since the grammar indicates all that a model could/should learn, whereas in the case of natural language, we must simply assume the data captures all grammatical rules.

The contributions of this work are as follows:
1. a theoretical contribution via the closed-form expression to compute a lower bound on perplexity for masked language modelling using the PCFG (based on pseudo-perplexity and the inside-outside algorithm).
2. a resource contribution via the synthetically generated corpus. The corpus contains 7.5M sentences generated using a state-split PCFG, learnt based on 500k sentences from the BookCorpus (part of the Pile). The authors examine the naturalness of their corpus by inspecting whether Zipf’s law applies, inspecting sentence lengths and inspecting frequencies of n-grams. In the process, they show that the state-split PCFG indeed leads to more natural corpora compared to a raw PCFG.
3. an evaluation of how models behave when trained on this corpus. 2 autoregressive models and 3 bidirectional models are trained on the data, their perplexity is compared to the lower bound, and correlations are reported comparing their probabilities to the PCFG’s probabilities. The key findings are: 1) causal language modelling has a higher perplexity than MLM, 2) lower perplexities correspond to higher correlations to the true distribution. Furthermore, the influence of corpus and model size are discussed.
4. an analysis of how POS tags are learnt over the course of training, and over the course of different layers, where the LMs’ performance is contrasted with the true distribution again.

**Questions For The Authors:**

1. Would you have idea of how the relative ordering of models would change when training them on 7.5M natural sentences from The Pile / BookCorpus?
2. Could you elaborate on how you would use your PCFG to execute an experiment like the one you mentioned in the introduction?
3. Could you give some examples of natural linguistic phenomena that your PCFG couldn't possibly capture?

**Reasons To Accept:**

- The paper presents an innovative approach to examining the capabilities of language models, namely by using synthetically generated data at a much larger scale than previous work. In addition to contributing the resulting PCFG and dataset as a resource, they contribute practical techniques required to make that work, and that approach as a whole could be adopted by future work.
- The paper presents a theoretical contribution to computing perplexity using a PCFG, assuming a masked language modelling setup. UPDATE AFTER REBUTTAL: as fellow reviewer ZP1G pointed out, there has been related work that made a similar contribution earlier this year.
- The paper presents a thorough analysis of how models behave on this synthetic corpus, and presents results that could not previously be computed because the true underlying distribution of natural language is unknown. These results include indications of how close to the lower bound on perplexity the LMs can get and to what extent the probabilities assigned to sentences (or words of certain POS tags) correlate with the probabilities of the true distribution.

Overall, I find this a refreshing take on language modelling: instead of approaching it as a black-box issue, it became fully transparent, without it being a toy setup.

**Reasons To Reject:**

- Even though the authors comment on the fact that PCFGs do not suffice for capturing *natural* linguistic variation in the limitations section, they do not actively address this in the main paper when analysing models (they only address this based on the statistics regarding naturalness). I would expect: 1) some type of comparative analysis illustrating how perplexities on 7.5M sentences from the natural corpus compare to the perplexities reported for the artificial corpus (to understand whether the better models in this setup are also the better models in a natural scenario). After all, you don't need the lower bound to see that some models perform better than others. 2) a discussion of what natural linguistic phenomena cannot be captured using this type of state-split PCFG. (after all, natural language is not context free...)
- I thoroughly enjoyed the work and recommend it for publication, but the reader is left wondering how this can be used in the future. The authors argue that this can be a tool for interpretability research, but in the paper itself that analysis remains somewhat shallow (the training dynamics of POS tags are much more simplistic than the example provided in the introduction regarding the prepositional dative). Given that the PCFG cannot really capture the intricacies of natural language, wouldn't we just be examining LMs' capabilities to learn extraordinarily large PCFGs, rather than *natural* language? Moreover, wouldn't the PCFG learnt itself be extremely hard to interpret?

**Reproducibility:**

4: Could mostly reproduce the results, but there may be some variation because of sample variance or minor variations in their interpretation of the protocol or method.

**Reviewer Confidence:**

3: Pretty sure, but there's a chance I missed something. Although I have a good feel for this area in general, I did not carefully check the paper's details, e.g., the math, experimental design, or novelty.

**Typos Grammar Style And Presentation Improvements:**

- line 11: "us define" --> "us to define"
- line 78: Misses an article perhaps? the / an exact lower bound
- line 86: "both masked language modelling" and... is something missing here? The usage of "both" suggests 2 things.
- line 208: Looks like the 2nd w has a typo in the subscript? <i should both be low/small?
- line 394: While the reader will understand what 3b means, the figure itself doesn't have a & b. Perhaps use the subcaption packages to actually have a & b in the figure itself, such that you can properly refer to them separately. This also applies to remaining figures and corresponding subfigures in the paper.
- It appears that you use the wrong convention for indicating numbers >999, you should use a comma instead of a period in English. (This applies to numbers >999 throughout the paper,  which is why I’m not including a line number here)
- line 588: & → and

---

> ### Author Rebuttal · Authors · 2023-08-29
>
> We thank the reviewer for their thoughtful review; it is highly encouraging to hear they find our approach ‘a refreshing take on language modelling’!
>
> A full analysis of how this paper can in turn be a tool in interpretability research would easily fill up one or several new papers (future work!). The key point here is that research on the faithfulness of interpretability methods, implicitly or explicitly, relies on the assumption that their usefulness/faithfulness does not depend on the nature of the source distribution. To the extent that that assumption holds, our work can be used to more thoroughly assess existing methods. But it can also be used to investigate the validity of that assumption itself.
>
> Q1: That is a great question that we did explore during experimentation, but ultimately left out for space reasons. One finding is that a masked LM, using the DeBERTa architecture that we also used in experiments reported in the paper, resulted in a considerably lower perplexity when trained on the original treebank, than when trained on the PCFG data. This shows that there are various linguistic signals in the original data that are not tackled by the PCFG, but that are exploited by the LM.
>
> This also relates strongly to Question 3. We suspect that most of the phenomena that the PCFG still fails to capture are related to semantic relations between phrases. While the state split procedure improves semantic relatedness with respect to the raw PCFG (as shown in Fig. 3b), the grammar is still not restrictive enough to only permit the generation of sentences that are semantically congruent across the entire sentence. In other words: our state-split PCFG overgenerates with respect to the original corpus, which increases the uncertainty over the generated corpus and results in a higher LM perplexity.
>
> While this issue could be improved with a better Treebank (see also our response to reviewer ybVx), there ultimately are limits to the expressiveness of a state split PCFG due to the finite number of non-terminal states. Simple example: to encode that a ‘dog’ is more likely to ‘bark’, and a ‘cat’ more likely to ‘meow’, we’d need separate S -> NP VP rules to carry over this information. The more fine-grained we want this semantic information to be encoded, the larger our PCFG would need to become.
>
> Thanks for these questions. Your curiosity about these aspects of the paper is an argument for us to incorporate a broader discussion in the paper. In the revised version we will incorporate the original Treebank perplexities as well, since it is a useful way of placing the performance of our PCFG LMs in context.
>
> Q2: The PCFG provides explicit access to the underlying hierarchical structure of the data that an LM needs to model. This allows us to investigate and control the distribution of particular linguistic constructions, since we have access to the rules that underlie these constructions. An investigation of such nature would be similar to the work of White and Cotterell (2021), in which they modify various typological features of artificial languages.

---

### Official Review · Reviewer_ybVx · 2023-08-05

**Soundness:** 4

**Excitement:**

4: Strong: This paper deepens the understanding of some phenomenon or lowers the barriers to an existing research direction.

**Missing References:**

- L185: The correct citation of _perplexity_ is [Jelinek+77] listed below.
 - L215: This method of getting a probability distribution for tokens under a PCFG has been extensively used in [Shin+21].

### References
 - [Jelinek+77]: F Jelinek, R L Mercer, L R Bahl, J K Baker. 1977. Perplexity—a measure of the difficulty of speech recognition tasks. J Acoust Soc Am 62, S63.
 - [Shin+21]: Richard Shin, Christopher Lin, Sam Thomson, Charles Chen, Subhro Roy, Emmanouil Antonios Platanios, Adam Pauls, Dan Klein, Jason Eisner, and Benjamin Van Durme. 2021. Constrained Language Models Yield Few-Shot Semantic Parsers. In Proceedings of the 2021 Conference on Empirical Methods in Natural Language Processing, pages 7699–7715, Online and Punta Cana, Dominican Republic. Association for Computational Linguistics.

**Paper Topic And Main Contributions:**

This paper presents a setup to evaluate and interpret neural language models using artificial data drawn from a language model learned from a PCFG. The PCFG is constructed using the state-split method over a large corpus. This method allows the definition for a closed-form expression for obtainable perplexity.

**Questions For The Authors:**

- L240: I love the notation of $\mathrm{\psi LL}$ for **ps**eudo-log-likelihood.
 - L292: You used a machine with 32GB RAM. In modern computer architectures (e.g. an EC2 machine on AWS), one can easily obtain a compute instance with RAM >256GB. I wonder what the PCFG would look like if induced over a much larger corpus (e.g. 10M sentences instead of 500k).
 - L296: _X-bar style_ grammar ensuring each rule has at most 2 children: would the term _Chomsky normal form_ be a better description of the constraints you put on the grammar? _X-bar_ requires the structure of NP-bar, NP, and N, and it contains nuisances such as specifiers (e.g. SpecNP).
 - Table 1: For the masked LMs, is the PPL column actually $\mathrm{\psi}LL$?

**Reasons To Accept:**

- A nice revival of the Berkeley parser and the construction of PCFGs using the state-split method.
 - A deep discussion on the use of a probabilistic grammar to evaluate neural sequence models.

**Reasons To Reject:**

- A fundamental problem is how much an induced PCFG actually resembles the distribution of natural language sequences. The authors conceded that PCFGs are not the only way to obtain a generative model over sequences. A decoder-based Transformer itself is also a generative model over sequences. One could in theory evaluate PCFGs under the distribution from a Transformer. The choice of PCFGs as the base distribution is not convincing in my opinion.

**Reproducibility:**

4: Could mostly reproduce the results, but there may be some variation because of sample variance or minor variations in their interpretation of the protocol or method.

**Reviewer Confidence:**

3: Pretty sure, but there's a chance I missed something. Although I have a good feel for this area in general, I did not carefully check the paper's details, e.g., the math, experimental design, or novelty.

---

> ### Author Rebuttal · Authors · 2023-08-29
>
> We thank the reviewer for the enthusiastic review. We will incorporate the suggested references in the paper. The work of Shin et al. is something we found out about after submitting, but is definitely highly relevant to our work: this goes to show that our approach is not just relevant to interpretability of language models, but other domains such as semantic parsing as well.
>
> We completely agree, of course, that the induced PCFGs are far from perfect approximations of natural language distributions (we're not trying to reverse the neural language model revolution!). Our key point is that there are many interesting distributions "in between" the idealized, simple distributions used in prior formal work, and the natural, complex distributions used in current empirical work. Our paper involves a considerable amount of nontrivial, technical work to provide one such intermediate distribution.
>
> Q1: Inducing the grammar on a larger language sample would probably get rid of some outlier rules, but we do not think that the current treebank size is the main limiting factor for the quality of the generated corpus. Instead of increasing the sample size, improving the quality of the original treebank is probably a more useful endeavor. This is something we definitely aim to explore in future work (and we would love to hear suggestions for high quality treebanks; it turned out to be challenging to find large-scale treebanks that provide a broad reflection of natural language).
>
> Q2: The X-Bar transformation is a procedure stemming from the implementation of Petrov et al. (2006), and does include NP-bar-like non-terminals. Since our final grammar also includes non-terminal rules of the form A -> B, it is not fully in Chomsky Normal Form.
>
> Q3: The PPL column for MLM is indeed ψLL, we’ll update the table.

---

### Meta-Review · Area_Chair_tbRG · 2023-09-19

**Recommendation:** 2

**Metareview:**

This paper evaluates and interprets neural language models using artificial data drawn from a language model learned from a PCFG. The PCFG is constructed using the state-split method over a large corpus. This method allows the definition for a closed-form expression for obtainable perplexity.

All reviewers bring up issues with using a PCFG to evaluate language models, questioning the claimed gains in interpretability and the degree of approximation of natural language. Despite a lively author-reviewer discussion period, there was no consensus among reviewers about the soundness and excitement of the work. The authors make some good arguments in their responses, and the manuscript can be strengthened by incorporating feedback from the author-reviewer discussion and expanding on the motivation for using a PCFG. On the positive side, reviewers praised the thorough experimental analysis and noted that the finding that causal and masked LMs show differences in their correlation with the true probabilities from the PCFG is quite interesting.

---

### Decision · Program_Chairs · 2023-10-07

**Decision:**

Accept-Findings

**Comment:**

This paper evaluates and interprets neural language models using artificial data drawn from a language model learned from a PCFG. The PCFG is constructed using the state-split method over a large corpus. This method allows the definition for a closed-form expression for obtainable perplexity.

All reviewers bring up issues with using a PCFG to evaluate language models, questioning the claimed gains in interpretability and the degree of approximation of natural language. Despite a lively author-reviewer discussion period, there was no consensus among reviewers about the soundness and excitement of the work. The authors make some good arguments in their responses, and the manuscript can be strengthened by incorporating feedback from the author-reviewer discussion and expanding on the motivation for using a PCFG. On the positive side, reviewers praised the thorough experimental analysis and noted that the finding that causal and masked LMs show differences in their correlation with the true probabilities from the PCFG is quite interesting.